# MOL-R1: TOWARDS EXPLICIT LONG-CoT REASONING IN MOLECULE DISCOVERY

## ABSTRACT

Large language models (LLMs), especially Explicit Long Chain-of-Thought (CoT) reasoning models like DeepSeek-R1 and QWQ, have demonstrated powerful reasoning capabilities, achieving impressive performance in commonsense reasoning and mathematical inference. Despite their effectiveness, Long-CoT reasoning models are often criticized for their limited ability and low efficiency in knowledge-intensive domains such as molecule discovery. Success in this field requires a precise understanding of domain knowledge, including molecular structures and chemical principles, which is challenging due to the inherent complexity of molecular data and the scarcity of high-quality expert annotations. To bridge this gap, we introduce **Mol-R1**, a novel framework designed to improve explainability and reasoning performance of R1-like Explicit Long-CoT Reasoning LLMs for molecule discovery. Our approach begins with a high-quality molecule reasoning generation dataset curated through **P**rior **R**egulation via **I**n-context **D**istillation (**PRID**), a dedicated distillation strategy to effectively generate paired reasoning traces guided by prior regulations. Building upon this, we introduce **Mo**lecular **I**terative **A**daptation (**MoIA**), a sophisticated training strategy that iteratively optimize the reasoning performance of R1-like reasoning LLMs. Finally, we examine the performance of Mol-R1 in a novel task, text-based molecule reasoning generation, showing superior performance against existing baselines.

## 1 INTRODUCTION

Molecule discovery focuses on finding new compounds with desired properties, essential for advances in drug development (Jayatunga et al., 2022), materials science (Higuchi et al., 2023), and sustainable chemistry (Garti et al., 2003). However, the vastness and complexity of chemical space make this process challenging. Traditional methods rely on fixed rules or templates (Walters & Barzilay, 2020), making them costly, time-consuming, and limited in creativity.

Large language models (LLMs) offer a promising alternative, allowing chemists to explore chemical space using natural language and symbolic notations (Edwards et al., 2021). Thanks to the large training corpora, LLMs can reason, follow instructions, and learn in context, enabling more flexible and efficient text-guided molecular discovery. For example, MolT5 (Edwards et al., 2022) introduced a molecule-caption translation

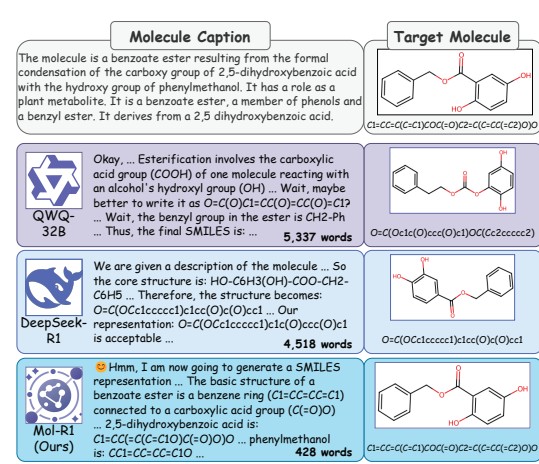

Figure 1: Derivations of a SMILES representation based on its caption. We compare the reasoning traces and final predictions of QWQ-32B (Team-Qwen, 2025), DeepSeek-R1 (Guo et al., 2025), and Our Mol-R1.

approach using the ChEBI-20 dataset, aligning SMILES strings (Weininger, 1988) with natural language descriptions. This allows chemists to design molecules simply by describing their requirements, while LLMs can then translate them into specific molecular structures (Li et al., 2024).

Although LLM-based approaches are effective, most of them follow a translation paradigm that focuses a one-to-one, direct mapping, which lacks intermediate reasoning steps and interpretability to support their results. However, in high-stakes real-world applications such as drug development, it is paramount to verify the model's reasoning process for building trust and informing critical decisions. The black-box nature of existing models not only obscures this verification process but also introduces unacceptable risks, as it can lead to the oversight of critical safety attributes, such as unforeseen toxicity or adverse effects.

Recently, DeepSeek-R1 (Guo et al., 2025) has emerged as a breakthrough model, excelling in mathematical inference and commonsense reasoning. By introducing explicit Long-CoT reasoning, R1-like reasoning models transform molecular discovery from a black-box translation task into a transparent, step-by-step reasoning generation process, thereby enabling the elucidation of structure-property relationships with human-interpretable justifications and possibly better performance as proved in Appendix B.

However, enabling R1-like reasoning in molecule discovery is challenging due to the lack of regulations: **(i) Cold-start data bottleneck (Pre-regulation)**: R1-like models requires a small, high-quality dataset of reasoning traces for initial adaptation, which seamlessly integrates step-by-step logical paths with its corresponding molecular structure and properties. However, existing datasets lack such traces, and manual annotation is prohibitively expensive due to the need for expert chemical knowledge. Similarly, automated methods like rejection sampling (Muennighoff et al., 2025) prove inefficient, as they lack the necessary chemical priors to guide the generation of valid reasoning chains. **(ii) Unconstrained exploration and error accumulation in reinforced policy optimization (Post-regulation)**: Unlike supervised learning, which provides a ground-truth signal at each step, reinforced policy optimization lacks deterministic knowledge supervision during the reasoning process. Consequently, the model is free to explore a vast problem space, often generating chemically invalid or logically unsound reasoning pathways. However, erroneous reasoning can coincidentally lead to a valid-looking final output and receive a high reward, thereby reinforcing the flawed logic that produced it. Ultimately, the model rapidly plateaus at superficial pattern recognition, unable to evolve beyond its initial knowledge base or discover novel, rule-compliant molecular structures.

To address the above challenges, we introduce **Mol-R1**, a novel R1-like explicit Long-CoT reasoning framework tailored for molecule discovery, enabling LLMs to mimic the deliberation process of chemists to generate complex molecule structures. Specifically, Mol-R1 aims to address two fundamental research questions: (i) How to efficiently generate high-quality, expert-aligned reasoning traces for **pre-regulation** using minimal expert annotations? and (ii) How could we effectively leverage these reasoning traces to **post-regulate** LLMs, enhancing molecular reasoning capability while ensuring training stability? Our main contributions are summarized as follows:

- We introduce Mol-R1, a novel framework to improve explainability of molecule discovery by enabling LLMs to mimic the deliberation of chemists.
- Mol-R1 began with a novel distillation strategy, PRID, to curate a unique reasoning dataset with a human-labelled in-context example and logic regulations.
- Leveraging the cold-start dataset, we introduced MoIA, which iteratively combines the determined supervision with policy-based rewards, to continually improve the performance of Mol-R1 for molecule reasoning generation.
- Comprehensive experiments validate the effectiveness of Mol-R1, demonstrating its significant potential to enable more explainable and chemist-like reasoning in molecule discovery.

## 2 RELATED WORK

### 2.1 LONG-COT REASONING WITH LLMS

With the scale of parameters, LLMs have shown powerful reasoning capability. The introduction of chain-of-thought (CoT) (Wei et al., 2022) has brought a revolutionary change to the utilization of LLMs' reasoning capabilities. CoT enables models to conduct more in-depth and systematic reasoning, thereby obtaining more reliable and accurate answers, which has laid a foundation for a series of subsequent methods that guide LLMs to reason. What's more, methods such as Tree of Thought (ToT) (Yao et al., 2023) and Graph of Thought (GoT) (Besta et al., 2024) have emerged. These methods construct tree or graph structures to explore problem spaces, enabling models to

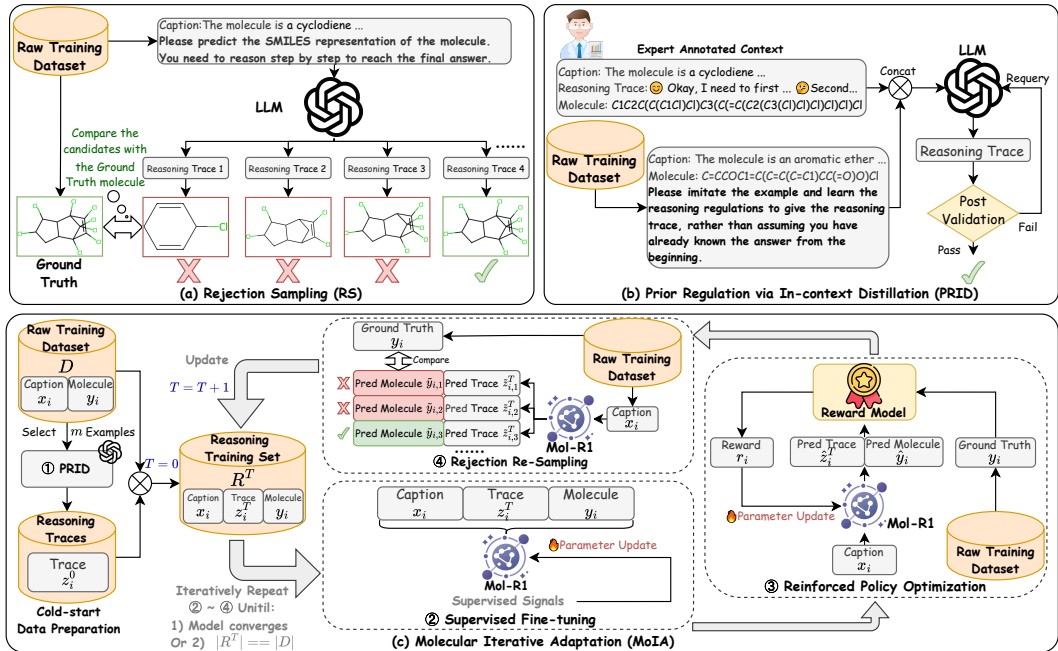

Figure 2: The overall framework of Mol-R1. (a) Applying Rejection Sampling to distill reasoning traces. (b) Using Prior Regulation via In-context Distillation to yield cold-start reasoning traces. (c) The pipeline of Molecular Iterative Adaptation (MoIA).

trace different reasoning paths and thereby find better solutions to complex problems. Furthermore, OpenAI's o1 (Jaech et al., 2024) and Deepseek's R1 (Guo et al., 2025), among others, made further innovations in this regard. They entrusted the exploration and decision-making of the Long-CoT reasoning process to the LLMs themselves through reinforcement learning (RL), which enhanced the reasoning capabilities of models, enabling them to make more flexible and intelligent judgments when facing complex problems.

## 2.2 LLMS FOR MOLECULE DISCOVERY

LLMs have demonstrated great potential in molecule discovery, encompassing areas such as molecular understanding, optimization, and generation (Edwards et al., 2021; Li et al., 2024). To bridge the gap between molecular representations and natural language texts, MolT5 (Edwards et al., 2022) first proposed the molecule-caption translation task, which contains pairs of molecule SMILES representations and their textual descriptions detailing structural patterns and chemical properties. However, existing works primarily focus on learning a one-to-one, direct mapping from a molecule to a static description (Li et al., 2025; Su et al., 2022). This paradigm, while effective for captioning, lacks the ability to perform dynamic reasoning or explain the underlying chemical principles. In contrast, our work introduces **text-based molecule reasoning generation**, instead of simply translating a molecule into a pre-defined description, to ensure a step-by-step explainable reasoning process in response to a specific query about the molecule.

## 3 PROBLEM FORMULATION

### 3.1 NOTATION

Considering the molecule caption as the query $x$, the corresponding molecule SMILES string as the output $y$, and the LLM parameters as $\theta$, we could define a conditional distribution over the output tokens: $\pi_\theta(y_t|x, y_{<t})$.

Given a training set $D = \{(x_i, y_i)\}_{i=1}^N$ with $N$ molecule SMILES strings and description pairs, the loss function for learning the policy $\pi_\theta$ via vanilla supervised fine-tuning can be denoted as:

$$\mathcal{L}(\theta) = \sum_{i=1}^N \left[ -\sum_{t=1}^{|y_i|} \log \pi_\theta(y_{i;t}|x_i, y_{i;<t}) \right], \tag{1}$$

where $|y_i|$ is the length of the $i$-th molecule SMILES string.

### 3.2 R1-LIKE EXPLICIT LONG-COT REASONING MODELS

Explicit Long-CoT reasoning models involve complex reasoning traces before giving the final answer, enabling them to explore and handle complex problems (Chen et al., 2025). Normally, explicit Long-CoT reasoning models follow a strict output format, enclosing their reasoning traces between two special tokens `<think>` and `</think>`, while the final answer will be placed between `<answer>` and `</answer>`. Compared to the previous CoT method (Wei et al., 2022) that utilizes prompts like `Let's think step by step`, R1-like explicit Long-CoT reasoning models are specifically trained and optimized for reasoning tasks, which encourages them to have a deeper, broader, and more reflective reasoning process before yielding the final answer.

### 3.3 TEXT-BASED MOLECULE REASONING GENERATION

In this work, we formalize the text-based molecule generation as a reasoning task. Different from directly learning the policy $\pi_\theta$, molecule reasoning generation requires LLMs to first generate their reasoning traces and then infer the final molecule SMILES string based on both the input query and their reasoning process. Therefore, given a molecule description $x$, the reasoning process $z$, and the corresponding molecule SMILES string $y$, the molecule reasoning involves the following two-stage decoding process with the same LLM parameterized by $\hat{\theta}$:

**Reasoning Process Generation.** First, we define a conditional distribution over the reasoning process tokens: $p_{\hat{\theta}}(z_t|x, z_{<t})$. In this stage, the LLM tries to utilize its chemical knowledge to interpret the descriptions into viable chemical structures.

**Final Answer Generation.** Next, the LLM generates the final SMILES string based on both the input description $x$ and its reasoning process $z$, with the conditional distribution: $q_{\hat{\theta}}(y_t|x, z, y_{<t})$.

In this case, molecule reasoning generation requires a special reasoning training set $R = \{(x_i, z_i, y_i)\}$ (where $(x_i, y_i) \in D$) to initialize the policies $q$ and $p$ via supervised fine-tuning (i.e., cold-start training, CS). Similarly, the objective for supervised fine-tuning can be denoted as:

$$\mathcal{L}(\theta) = -\sum_{i=1}^{|R|} \left( \sum_{t=1}^{|z_i|} \log p_\theta(z_{i,t}|x_i, z_{i,<t}) + \sum_{t=1}^{|y_i|} \log q_\theta(y_{i,t}|x_i, z_i, y_{i,<t}) \right) \tag{2}$$

where $|z_i|$ and $|y_i|$ are the length of the reasoning process and molecule SMILES string, respectively, while $|R|$ denotes the number of examples in $R$.

Text-based molecule reasoning generation simulates the problem-solving process of human experts by progressively analyzing the captions and inferring potential molecule structures, which could potentially enhance the explainability of the final predictions.

## 4 THE FRAMEWORK OF MOL-R1

In this section, we introduce the framework of Mol-R1. As shown in Figure 2, Mol-R1 consists of two important components, Prior Regulation via In-context Distillation (**PRID**) and Molecular Iterative Adaptation (**MoIA**). We first detail PRID, which targets the post-hoc regulation of the cold-start data. Subsequently, we present the design of MoIA, which iteratively involves dynamic interactions between supervised signals and policy-based rewards.

### 4.1 PRIOR REGULATION VIA IN-CONTEXT DISTILLATION

Facilitating explicit Long-CoT for text-based molecule reasoning generation initially depends on a small yet high-quality reasoning dataset for cold-start training, one that seamlessly integrates reasoning traces with the corresponding captions and molecules. However, the curation of the cold-start reasoning training set of Mol-R1 mainly faces the following challenges:

1. Molecule-caption datasets, such as ChEBI-20 (Edwards et al., 2022), lack the crucial reasoning traces that link captions with molecules. Annotating such reasoning traces is often costly and time-consuming due to the inherent complexity of molecular structure and the demand for highly specialized knowledge from domain experts.

2. Existing LLMs often fail to infer correct molecules through rejection sampling, due to a lack of molecular knowledge, consequently lowering the sampling efficiency.

3. The reasoning traces distilled from existing explicit Long-CoT reasoning models via rejection sampling tend to broadly explore the reasoning space. However, molecular reasoning demands precise knowledge and rigorous logic, necessitating a careful balance between the breadth of exploration and the precision of knowledge.

To overcome these challenges, we propose Prior Regulation via In-context Distillation for cold-start reasoning trace distillation. Guided by the high-quality reasoning example in PRID, LLM can learn from the inherent logic and regulations within the reasoning process. The detailed pipeline is demonstrated in Figure 2 (b):

**Firstly**, a human expert writes **only one** high-quality reasoning trace example $c$ with prior regulations for text-based molecule generation (See in Appendix E.1), which focuses on interpreting potential molecular structures from the descriptions and guiding the reasoning process of prior regulations.

**Secondly**, we perform in-context distillation based on the expert example. Here, expert annotation could serve as a guideline to prompt LLMs to encapsulate the underlying logical patterns and reasoning principles, providing soft boundaries for the reasoning process. Our approach strategically leverages a rather small subset of $m(m \ll N)$ examples from the raw training set $D$, while preserving free exploration of the remaining space.

**Thirdly**, we introduce a post-validation stage to examine the quality of reasoning traces. If the reasoning trace fails to pass the evaluation of the post-validator, we will proceed with a re-query. Specifically, we implement a format validator and a score validator. The former is to ensure that the reasoning traces are enclosed between `<think>` and `</think>`, while the latter will be performed by adopting the LLM as a judge to score the generated reasoning trace based on the precision of the knowledge and logical coherence.

## 4.2 MoIA: MOLECULAR ITERATIVE ADAPTATION

Training explicit Long-CoT reasoning models typically begins with an initial cold-start supervised fine-tuning stage, which introduces foundational, deterministic knowledge and establishes a consistent format for reasoning traces. Following the cold-start SFT, the model undergoes Reinforced Policy Optimization (RPO) on a larger amount of data without reasoning traces, which enables LLMs to learn the molecule SMILES syntax via policy-based rewards.

However, we observe that in the previous training pipeline, the reasoning model performance will quickly converge during the RPO stage, indicating a knowledge bottleneck. Therefore, to achieve better results, we need to incorporate more deterministic or established knowledge supervision into the training process.

In this case, we propose MoIA, an iterative training framework that iteratively involves dynamic interactions between supervised signals and policy-based rewards. Specifically, we define the iteration of MoIA as $T$. We will iteratively update the reasoning training set $R^T$, the reasoning traces $z_i^T$ and optimize the model parameters $\theta^T$. Figure 2 (c) demonstrates the detailed pipeline of MoIA:

① As discussed previously, we conduct the PRID process for cold-start data preparation, based on the prior guidance from expert annotated example $c$:

$$z_i^0 = PRID(x_i, y_i|c), \tag{3}$$

where the superscript in $z_i^0$ refers to the initial iteration of MoIA ($T = 0$). Then, the distilled reasoning traces will form the reasoning training set $R^T$ at the initial iteration:

$$R^T = \{(x_i, z_i^T, y_i)\}, \tag{4}$$

② We then introduce deterministic knowledge and perform supervised fine-tuning on the reasoning training set $R^T$ by minimizing the autoregressive loss as mentioned in Equation (2), thereby updating the model parameters $\theta^T \to \hat{\theta}^T$:

$$\hat{\theta}^T = \underset{\theta}{argmin}\mathcal{L}(\theta), \tag{5}$$

| Method | BLEU↑ | EM↑ | Levenshtein↓ | MACCS FTS↑ | RDK FTS↑ | Morgan FTS↑ | Validity↑ |
|---|---|---|---|---|---|---|---|
| **Baselines (w/ Reasoning)** | | | | | | | |
| Llama3-8B-R1-distilled | 0.097 | 0.011 | 64.62 | 0.442 | 0.270 | 0.213 | 0.228 |
| QWQ-32B | 0.181 | 0.032 | 62.89 | 0.584 | 0.379 | 0.331 | 0.518 |
| DeepSeek-R1 | 0.031 | 0.156 | 68.47 | 0.732 | 0.600 | 0.580 | 0.522 |
| GPT-oss-120b | 0.368 | 0.128 | 50.40 | 0.823 | 0.650 | 0.579 | 0.748 |
| **Base Model SFT (w/o Reasoning)** | | | | | | | |
| Llama3.1-8B (n=1053) | 0.375 | 0.021 | 95.34 | 0.676 | 0.502 | 0.416 | 0.612 |
| **Mol-R1 Cold-start Configurations (w/ Reasoning)** | | | | | | | |
| Mol-R1-RS-G (n=2943) | 0.179 | 0.047 | 56.42 | 0.623 | 0.385 | 0.329 | 0.701 |
| Mol-R1-PRID-G (n=1053) | 0.605 | 0.015 | 40.08 | 0.718 | 0.503 | 0.417 | 0.675 |
| Mol-R1-PRID-4o (n=1053) | 0.636 | 0.038 | 39.67 | 0.761 | 0.549 | 0.475 | 0.750 |
| **Mol-R1 MoIA Iterations(w/ Reasoning)** | | | | | | | |
| Mol-R1 (T=0) | 0.650 | 0.144 | 34.33 | 0.804 | 0.639 | 0.558 | 0.819 |
| Mol-R1 (T=1) | 0.655 | 0.216 | 32.87 | 0.821 | 0.673 | 0.596 | 0.863 |
| Mol-R1 (T=2) | 0.641 | 0.234 | 32.94 | 0.823 | 0.684 | 0.612 | 0.847 |

Table 1: Overall performance comparison ( Best , Second Best ).

③ We conduct RPO on the raw training dataset $D$. For each instance $i$, we provide only the input caption $x_i$ to the LLM, allowing it to explore its reasoning trace $\hat{z}_i^T$ and final prediction $\hat{y}_i$. Subsequently, the complete generated output as well as the ground truth $y_i$ is fed into a reward model $RM$ to acquire a feedback reward $r_i$:

$$r_i = RM(\hat{z}_i^T, \hat{y}_i, y_i), \tag{6}$$

This acquired reward $r_i$ provides crucial feedback on the molecule generation policy and is then used to update the LLM's parameters from $\hat{\theta}^T$ to $\theta^{T+1}$ by maximizing the expected reward:

$$\theta^{T+1} = \underset{\theta}{\arg\max}\, \mathbb{E}_{(\hat{z}_i, \hat{y}_i) \sim P_\theta(\cdot|x_i)}[r_i], \tag{7}$$

where $\mathbb{E}_{(\hat{z}_i, \hat{y}_i) \sim P_\theta(\cdot|x_i)}[r_i]$ denotes the expected reward for the generated reasoning trace $\hat{z}_i^T$ and final prediction $\hat{y}_i$ sampled from the LLM's current policy $P$, given input $x_i$.

④ We propose rejection re-sampling to update and improve the quality of the reasoning training set for the next iteration. Using the newly optimized model parameters $\theta^{T+1}$, we generate multiple candidate reasoning traces $\tilde{z}_{i,k}^T$ and predictions $\tilde{y}_{i,k}$ for each instance $i$ in the raw training dataset $D$, where $k$ denotes the $k$-th rejection sampling attempt. Crucially, only those generated results where the final prediction $\tilde{y}_{i,k}$ is completely identical to the ground truth $y_i$ are accepted. Therefore, we have:

$$R^{T+1} = \bigcup_{i \in D} \left\{ (x_i, \tilde{z}_{i,k}^T, y_i) \mid \exists k : (\tilde{z}_{i,k}^T, \tilde{y}_{i,k}) \sim P_{\theta^{T+1}}(\cdot|x_i) \wedge \tilde{y}_{i,k} = y_i \right\}, \tag{8}$$

where $y_i$ represents the ground truth molecule for input caption $x_i$, and only generated outputs $(\tilde{z}_{i,k}^T, \tilde{y}_{i,k})$ where the prediction $\tilde{y}_{i,k}$ perfectly matches this ground truth are included. The existential quantifier $\exists k$ : signifies that if, for a given $x_i$, at least one of the $k$ rejection sampling attempts yields a correct prediction, that specific successful triplet $(x_i, \tilde{z}_{i,k}^T, y_i)$, where $z_i^{T+1} = \tilde{z}_{i,k}^T$, is included in $R^{T+1}$. This process effectively curates and amplifies correct and deterministic knowledge within the accepted reasoning traces, thereby significantly enhancing the molecular understanding and overall reasoning capability of the LLM.

Finally, we let $T = T + 1$, and iteratively repeat the step ② - ④ until:

1) $\theta^T$ converges, i.e., the performance of Mol-R1 stabilizes;

2) $|R^T| == |D|$, i.e., the entire raw training dataset is fully annotated with reasoning traces.

# 5 EXPERIMENTS

## 5.1 IMPLEMENTATION DETAILS

To distillate reasoning traces for cold-start training, we mainly select GPT-4o (Hurst et al., 2024). Additionally, gemma-3-27b-it (abbreviated gemma3-27B) and gemma-3-12b-it (abbreviated gemma3-12B) (Team-Gemma et al., 2025) are adopted via rejection sampling for comparison. We utilize the vllm[1] and OpenAI[2] frameworks for querying the LLMs. In all, we sampled 1,053 reasoning

---

[1] https://blog.vllm.ai
[2] https://openai.com/api/

traces for the cold-start training of Mol-R1. Meanwhile, we merge the rejection sampling results of gemma3-27B and gemma3-12B, obtaining 2,943 valid reasoning traces. Meanwhile, we adopt Llama-3.1-8B-Instruct (abbreviated llama3.1-8B) (Dubey et al., 2024) as the model designated for MoIA training. Here, the llama-factory[3] is adopted for supervised fine-tuning, and the open-r1[4] framework and Group Relative Policy Optimization (GRPO) (Guo et al., 2025) are employed for the implementation of reinforced policy optimization. For more experimental details, please ref to Appendix A and C.

## 5.2 EVALUATION METRICS

We evaluate the performance of Mol-R1 from two distinct perspectives:

**Accuracy of Generated Molecules.** We adopt **BLEU**, Exact Match (**EM**), **Levenshtein** distance, **Validity**, and Molecule Fingerprints scores (specifically, **MACCS FTS**, **RDK FTS**, and **Morgan FTS**) to evaluate the similarity between the predictions and the ground truth.

**Quality of Reasoning Traces.** Effective reasoning traces inherently enhance the interpretability and trustworthiness of the generated molecule. Conversely, errors in the reasoning traces allow chemists to identify underlying flaws and preemptively assess the reliability of the final molecular answer. To evaluate the quality of reasoning traces, we use Gemma3-27B to mimic a chemist's judgment of the reasoning traces. The judge will predict whether the reasoning trace itself could lead to a correct conclusion solely based on its content and without access to the final answer. Then, we compare the judge's prediction against the actual correctness of the final answer. If the judge's prediction is the same to the actual correctness of the generated molecule, we define the reasoning trace as consistent. Finally, we report the F1 score of the consistency (i.e., **Consistent-F1**) to indicate the quality of generated reasoning traces. For more details, please refer to Appendix E.2.

## 5.3 RESULT & DISCUSSION

In this work, we specifically include four powerful explicit Long-CoT reasoning LLMs, Llama3-8B-R1-distilled (Guo et al., 2025), QWQ-32B (Team-Qwen, 2025), DeepSeek-R1 (Guo et al., 2025), and GPT-oss-120b (Agarwal et al., 2025), as baselines. Concurrently, we demonstrate the performance of the base model using the same molecule-caption pairs with the cold-start reasoning data to ablate the effectiveness of explicit reasoning traces. Furthermore, we report the cold-start performance of Mol-R1 under various cold-start configurations, as well as the performance across various MoIA iterations. Notably, Table 1 presents a comparison of molecule generation accuracy across the above models, while Figure 3 illustrates the Consistent-F1 scores of their respective reasoning traces.

Generally, Mol-R1 (T=2) demonstrates superior performance in molecule generation accuracy, achieving an EM score of 0.234 and the highest Molecule FTS metrics across all these models. When directly compared to even the most advanced explicit Long-CoT reasoning baselines, such as QWQ-32B, GPT-oss-120b, and DeepSeek-R1, Mol-R1 (T=2) significantly outperforms QWQ-32B with a BLEU score that is a staggering 354% higher, indicating a much more similar prediction to reference molecules. Furthermore, Mol-R1 (T=2) also obtains an EM score that is 1.5 times better than DeepSeek-R1, showcasing its enhanced ability to generate accurate molecules based on the text descriptions.

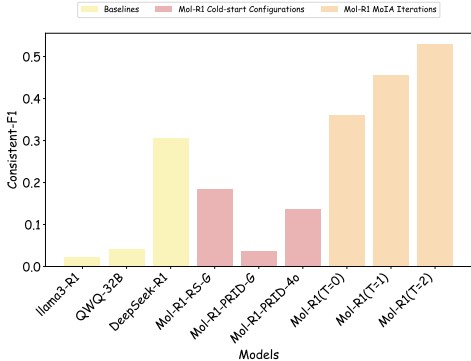

Figure 3: Comparison of reasoning trace quality across different models, where higher values of the Consistent-F1 score indicate more reliable and robust reasoning.

Beyond generation accuracy, Mol-R1 (T=2) also achieves the highest Consistent-F1 score for its reasoning trace quality. This indicates that in most cases, the accuracy of the predictions can be effectively pre-assessed by analyzing its reasoning traces. This highlights the exceptional explainability and robustness of Mol-R1.

---

[3] https://llamafactory.readthedocs.io
[4] https://open-r1.com/

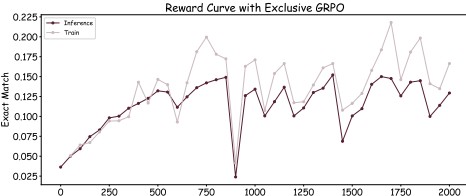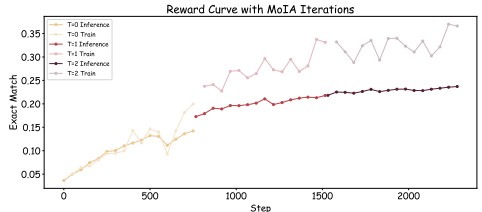

Figure 4: Training and inference dynamics of exclusive GRPO (left) and MoIA iterations (right).

Next, we focus on ablating the key components of Mol-R1 with extensive experiments: (i) Assessing the effectiveness of PRID in efficiently generating high-quality, expert-aligned reasoning traces; and (ii) the necessity of applying MoIA compared to exclusive policy optimization.

### 5.3.1 ASSESSMENT OF COLD-START CONFIGURATIONS

In this part, we prioritize the performance of Prior Regulation via In-context Distillation (PRID) by examining the impacts of different models and algorithms for distilling the cold-start reasoning data for Mol-R1 (i.e., cold-start configurations).

First of all, we test the supervised fine-tuning performance of the base model, Llama3.1-8B, with and without reasoning traces at the same data scale (i.e., n=1053). The results shown in Table 1 demonstrate that with the same 1,053 molecule-caption pairs, Mol-R1-PRID-4o, trained with reasoning traces distilled from GPT-4o via PRID, achieves a significant improvement over the Llama3.1-8B (the base model w/o reasoning). Notably, Mol-R1-PRID-4o achieves a BLEU score of 0.636 compared to 0.375 for Llama3.1-8B, and an exact match score of 0.038 versus 0.021, indicating substantial gains in molecule generation accuracy attributable to the inclusion of reasoning traces.

Then, we investigated the impacts of using a different LLM for PRID, introducing gemma3-12B for comparison. We observed that Mol-R1-PRID-G (using gemma3-12B) and Mol-R1-PRID-4o (using GPT-4o), both utilizing PRID for reasoning trace distillation, achieved comparable results in terms of molecule generation accuracy with close BLEU scores, and other metrics like Molecule FTS scores and Validity, also indicated in Table 1. This suggests that PRID is generally effective and robust across different LLMs.

However, when we consider the quality of reasoning traces, a notable divergence appears. Mol-R1-PRID-G achieves a significantly lower Consistent-F1 score compared to Mol-R1-PRID-4o. This indicates that while both models can generate molecules with similar accuracy, the reasoning traces produced by gemma3-12B are considerably less consistent or reliable for pre-assessing prediction accuracy. In essence, although gemma3-12B can contribute to effective molecule generation, the robustness and explainability of its underlying reasoning traces are not on par with GPT-4o. This highlights that while PRID is effective, its full potential is realized when paired with a more powerful LLM, underscoring the importance of model choice.

Finally, we compared PRID with Rejection Sampling (RS). Specifically, we evaluated Mol-R1-RS-G, which was trained on 2,943 reasoning traces distilled using gemma3-12B and gemma3-27B via Rejection Sampling. The results are also shown in Table 1. It can be observed that, despite being trained on a considerably larger dataset for cold-start (2,943 data points for RS vs. 1,053 for PRID), Mol-R1-RS-G still achieves inferior performance. For instance, Mol-R1-RS-G obtained a BLEU score of 0.179 and a Morgan FTS score of 0.329, which are even lower than those obtained by Mol-R1-PRID-G.

These findings suggest that for highly specialized domain tasks like text-based molecule reasoning generation, the distillation of reasoning traces requires specific regulations. Rejection Sampling, which allows LLMs to explore more freely, appears to introduce excessive noise and unnecessary reasoning paths into the reasoning process, which ultimately compromises training performance. In contrast, PRID, by incorporating the expert-annotated example and regulations, provides LLMs with robust guidance on how to reason effectively to produce the target molecule, which is crucial for improving the performance on text-based molecule reasoning generation.

### 5.3.2 ABLATION OF ITERATIVE TRAINING

Figure 4 illustrates the distinct training dynamics between exclusive GRPO and MoIA. For comparison, we initiated exclusive GRPO training immediately following the supervised fine-tuning at the first iteration (T=0) of MoIA.

A clear distinction emerges: the exact match (EM) reward for exclusive GRPO converges rapidly around 0.14 after approximately 800 steps. Critically, the training remains unstable with significant fluctuations throughout the remaining steps, indicating a premature plateau and lack of sustained improvement.

In contrast, within the MoIA framework, the exact match reward continues to improve and does not fully converge until roughly 2,000 steps of policy optimization. Meanwhile, as shown in Figure 3, the Consistent-F1 score continuously improves, reflecting a consistent enhancement in the quality of reasoning traces as MoIA iterations advance. This prolonged and stable improvement demonstrates how MoIA's iterative training strategy effectively solidifies deterministic knowledge, enabling the model to continuously self-evolve, which is crucial to improve both the reasoning capability and the accuracy of molecule generation.

### 5.4 CASE STUDY

Here, we present a detailed example, shown in Figure 5, to demonstrate the evolution of Mol-R1 across MoIA iterations. In this example, models need to identify the SMILES representation of "O-methylmalonylcarnitine" and then determine which carboxyl group (-COOH) loses a proton to form the monoanion, or which additional carboxyl proton is lost if the molecule has multiple acidic sites.

At the initial iteration (T=0), guided by the PRID mechanism, the model naturally began to reflect on its previous reasoning. However, the generated molecule still contained errors; while the overall structure was similar, the two hydroxyl groups were not fully ionized as required.

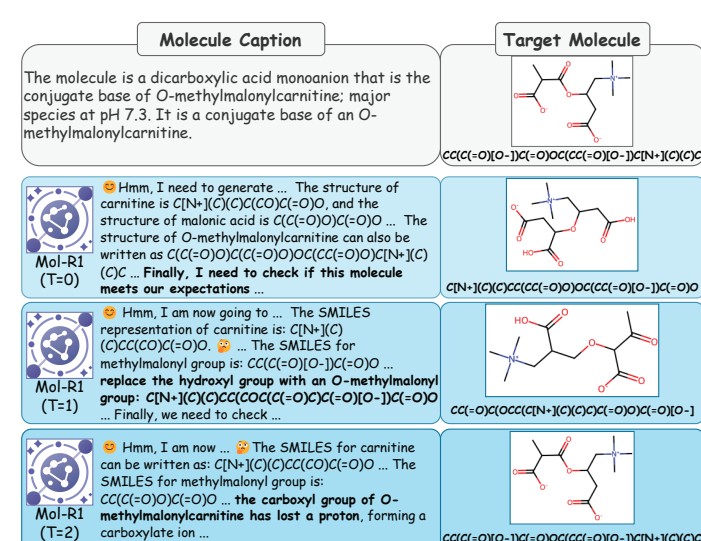

Figure 5: Case study of the reasoning traces and final predictions across three MoIA iterations (T=0, 1, 2) of Mol-R1.

Moving to the next iteration (T=1), the model recognized the need to modify or replace specific groups (e.g., handling the deprotonation of hydroxyl groups). Yet, this process remained incomplete, resulting in one of the hydroxyl groups not being fully ionized.

Finally, at MoIA (T=2), through iterative optimization and verification of its knowledge, the model accurately generated the SMILES representation that perfectly matched the target molecule. This progression clearly shows the model's ability to overcome complex molecular structure generation hurdles through MoIA.

## 6 CONCLUSION

In this work, we introduce Mol-R1, an innovative framework specifically designed to enhance explainability and reasoning performance of R1-like explicit Long-CoT reasoning LLMs in text-based molecule generation. Mol-R1 began with Prior Regulation via In-context Distillation (PRID), which effectively curated a high-quality reasoning dataset via in-context learning guided by an expert-annotated example, addressing the inherent complexities and knowledge scarcity of reasoning traces in molecule discovery, Building upon this, we then introduced Molecular Iterative Adaptation (MoIA), which iteratively combines supervised signals with policy-based rewards for post-regulation, enabling Mol-R1 to achieve superior performance in text-based molecule reasoning generation.

ETHICS STATEMENT

This work does not involve ethic issues.

REPRODICIBILITY STATEMENT

We will release all the codes and checkpoints once the paper gets published.

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

# A   IMPLEMENTATION DETAILS

In this section, we list extra implementation details that help readers fully understand our approach, including hyper-parameters, statistics of the cold-start data, and the algorithm selected for Reinforced Policy Optimization.

## A.1   HYPER-PARAMETERS

| Item | Value |
|---|---|
| **Supervised Fine-tuning (SFT)** | |
| gpu_number (A800) | 2 |
| per_device_train_batch_size | 1 |
| gradient_accumulation_steps | 4 |
| learning_rate | 1.0e-5 |
| num_train_epochs | 5 |
| lr_scheduler_type | cosine |
| warmup_ratio | 0.1 |
| **Reinforced Policy Optimization (RPO)** | |
| gpu_number (A800) | 8 |
| learning_rate | 1.0e-6 |
| weight_decay | 1.0e-2 |
| kl_coef | 1.0e-2 |
| n | 5 |
| rollout.temperature | 1.0 |
| global_batch_size | 128 |
| rollout_batch_size | 512 |
| micro_batch_size_per_device_for_update | 4 |
| **Inference** | |
| temperature | 0.6 |
| top_p | 0.9 |
| max_tokens | 10000 |

Table 2: Hyper-parameters

Here, we present the specific hyper-parameters that govern Mol-R1's training and inference processes, as summarized in Table 2. All computational experiments were performed on Nvidia A800 GPUs.

For Supervised Fine-tuning (SFT), we employed two A800 GPUs. The training configuration of SFT included a per_device_train_batch_size of 1 with gradient_accumulation_steps set to 4. We used a learning_rate of 1.0e-5, a cosine learning rate scheduler with a warmup_ratio of 0.1, and trained for 5 epochs.

During Reinforced Policy Optimization (RPO), eight A800 GPUs were utilized. Key parameters included a learning rate of 1.0e-6, weight_decay of 1.0e-2, and a kl_coef of 1.0e-2. The number for sampling n was set to 5, rollout.temperature to 1.0, global_batch_size to 128, rollout_batch_size to 512, and micro_batch_size_per_device_for_update to 4.

Finally, for Inference, the decoding strategy used a temperature of 0.6, top_p of 0.9, and a max_tokens limit of 10000 to control generation diversity and length.

## A.2   DATA STATISTICS

| Dataset | # Entries | # Average Length |
|---|---|---|
| **Raw Training Dataset** | | |
| ChEBI-20 | 26,407 | 58.73 |
| **Mol-R1 Cold-start Configurations** | | |
| RS-G | 2,943 | 258.03 |
| PRID-G | 1,053 | 870.67 |
| PRID-4o ($R^0$) | 1,053 | 893.88 |
| **Mol-R1 Reasoning Training Set** $R^T$ | | |
| $R^1$ | 7,285 | 763.24 |
| $R^2$ | 8,700 | 794.43 |

Table 3: Statistics of the supervised fine-tuning data.

This section details the statistics of the datasets used for cold-start supervised fine-tuning (i.e., MoIA T=0 SFT), encompassing our raw training data, cold-start data with various configurations, and the reasoning training sets at different MoIA iterations. Table 3 presents the number of entries and average length for each dataset.

### A.2.1 RAW TRAINING DATASET

We adopt ChEBI-20 as our foundational raw training dataset, comprising 26,407 entries with an average length of 58.73. This dataset contains molecule-caption pairs without reasoning traces.

### A.2.2 MOL-R1 COLD-START DATA WITH VARIOUS CONFIGURATIONS

These datasets are generated with different cold-start configurations utilized in the initial SFT phase of Mol-R1.

- **RS-G**: Generated by Gemma3-12B and Gemma3-27B via Rejection Sampling, containing 2,943 entries with an average length of 258.03.

- **PRID-G**: Features 1,053 entries, but with a significantly longer average length of 870.67, indicating these entries incorporate more extended reasoning sequences. The "G" here denotes that the traces were generated using the Gemma3-12B model via PRID.

- **PRID-4o**: Also known as $R^0$. It shares the same number of entries as PRID-G (1,053 entries) but has a slightly longer average length of 893.88. The "4o" signifies the use of the GPT-4o model for reasoning trace generation. A comparison between PRID-G and PRID-4o reveals minor differences in the level of detail within the reasoning traces generated by PRID with different models.

### A.2.3 MOL-R1 REASONING TRAINING SETS

These datasets were iteratively updated throughout the MoIA via Rejection Sampling, aimed at progressively refining the model's reasoning capabilities.

- $R^1$: The iteration T=1 of the reasoning training set includes 7,285 entries with an average length of 763.24.

- $R^2$: The iteration T=2 expands to 8,700 entries, with an average length of 794.43. This growth in both scale and average length across iterations reflects the increasing complexity and richness of the accumulated reasoning traces via MoIA.

### A.3 GROUP RELATIVE POLICY OPTMIZATION

Group Relative Policy Optimization (GRPO) is a reinforcement learning algorithm that aims to enhance the reasoning capabilities of LLMs and decision quality by revisiting and re-evaluating past reasoning steps (Guo et al., 2025). The core idea behind GRPO is to optimize the policy model using within-group relative rewards, rather than relying on a traditional critic model (Schulman et al., 2017). Specifically, GRPO samples a set of actions at each state (i.e., responses of LLMs with the current model parameters) and then updates the policy model based on the reward differences among these actions, which avoids the estimation of value functions, simplifying the algorithm while maintaining strong performance.

Mathematically, given a molecule description $x$, GRPO first samples a group of $G$ predicted molecules $\{m_1, m_2, ..., m_G\}$ from the old policy model $\pi_{\theta_{old}}$ and optimize the policy model $\pi_\theta$ via the following loss function:

$$\mathcal{L}(\theta) = \frac{1}{G} \sum_{k=1}^{G} \{\min[(\frac{\pi_\theta(m_k|x)}{\pi_{\theta_{old}}(m_k|x)})A_k,$$

$$clip(\frac{\pi_\theta(m_k|x)}{\pi_{\theta_{old}}(m_k|x)}, 1 - \epsilon, 1 + \epsilon)A_k]$$

$$- \beta \mathcal{D}_{KL}(\pi_\theta || \pi_{ref})\}, \tag{9}$$

$$\mathcal{D}_{KL}(\pi_\theta || \pi_{ref}) = \frac{\pi_{ref}(m_k|x)}{\pi_\theta(m_k|x)} - \log \frac{\pi_{ref}(m_k|x)}{\pi_\theta(m_k|x)} - 1, \tag{10}$$

where $\mathcal{D}_{KL}$ denotes the KL divergence between the policy model and the reference model, which enhances the policy stability and convergence, while $\beta$ and $\epsilon$ are hyper-parameters, and $A_k$ is the group advantage that calculated by a group of rewards $\{r_1, r_2, ...r_G\}$:

$$A_k = \frac{r_k - mean(\{r_1, r_2, ..., r_G\})}{std(\{r_1, r_2, ..., r_G\})} \tag{11}$$

The design of the reward function $r$ is also critical in GRPO, as it directly influences the quality and performance of the learned policy. In this work, we mainly adopt the **Exact Match** (**EM**) rewards for our policy training. The exact match score assesses whether the predicted molecule is identical to the ground truth. Specifically, we convert the generated SMILES string into standard form using RDKit[5], and then compare it with the ground truth.

four different reward functions:

- **Exact Match Rewards:** The exact match score assesses whether the predicted molecule is identical to the ground truth. Specifically, we convert the generated SMILES string into the standard form via RDKit, and then compare it with the ground truth.

- **Format Rewards:** The format reward requires LLMs to generate the reasoning process and the final answer following the pre-defined pattern. The reasoning process should be placed between the tags `<think>` and `</think>`, and the final answer should be placed between `<answer>` and `</answer>`.

- **Length Rewards:** The length rewards encourage LLMs to engage in deeper reasoning, exploring more potential paths and thinking more carefully. However, if the reasoning trace is too long, it could harm the output efficiency (Chiang & Lee, 2024). In this work, we set a manual threshold for these rewards to control the "thinking budget"—beyond a certain token length, the length reward no longer increases.

- **Similarity Rewards:** Unlike exact match rewards, similarity rewards are designed to be smoother. While accuracy rewards strictly require the generation of exactly matched molecules, similarity rewards can provide positive feedback for predictions that are merely similar.

In practice, these reward functions are typically combined rather than used independently. We explore the combinations of the reward functions through detailed experiments to optimize the reasoning performance of Mol-R1 in text-based molecule reasoning generation.

## B  INFORMATION-THEORETIC PROOF OF EXPLICIT REASONING EFFECTIVENESS

### 1. NOTATIONS AND DEFINITIONS

Let's define the random variables and information-theoretic concepts used:

- $Q$: Random variable representing the **problem**.
- $A$: Random variable representing the **true answer**. We assume a true conditional probability distribution $P(A|Q)$.
- $R$: Random variable representing the **reasoning path**.
- $A_{\text{out}}$: Random variable representing the **model's output answer**.

We utilize core information-theoretic definitions:

- **Entropy** $H(X)$: Measures the uncertainty of a random variable $X$. $H(X) = -\sum_x P(x) \log P(x)$.

---

[5]https://www.rdkit.org/

- **Conditional Entropy** $H(X|Y)$: Measures the uncertainty of $X$ given $Y$. $H(X|Y) = -\sum_{x,y} P(x,y) \log P(x|y)$.

- **Mutual Information** $I(X;Y)$: Quantifies the amount of information shared between $X$ and $Y$. $I(X;Y) = H(X) - H(X|Y) = H(Y) - H(Y|X) = H(X) + H(Y) - H(X,Y)$.

2. MODEL ARCHITECTURES

- **Direct Output Model** ($M_D$): Directly maps problem $Q$ to output answer $A_{\text{out}}$. This can be viewed as a conditional probability distribution $P(A_{\text{out}}|Q)$.

- **Explicit Reasoning Model** ($M_E$): Comprises two stages:

    1. **Reasoning Path Generator** ($P_{\text{path}}$): Generates a reasoning path $R$ from $Q$. Represented by $P(R|Q)$.

    2. **Result Generator** ($G_{\text{res}}$): Generates the final answer $A_{\text{out}}$ from the reasoning path $R$. Represented by $P(A_{\text{out}}|R)$.

    The overall process for $M_E$ is given by $P(A_{\text{out}}|Q) = \sum_R P(A_{\text{out}}|R)P(R|Q)$.

4. PROOF DERIVATION

4.1. IDEAL EXPLICIT REASONING PATH: ENTROPY REDUCTION AND INFORMATION GAIN

**Premise**: Assume an **ideal and true reasoning path ($R_{\text{true}}$)** exists. This path perfectly captures all relevant information about the answer $A$ from the problem $Q$ without introducing noise. In this ideal scenario, $Q$, $R_{\text{true}}$, and $A$ form a **Markov chain**: $Q \to R_{\text{true}} \to A$. This means that given $R_{\text{true}}$, all information about $A$ is contained within $R_{\text{true}}$, and $Q$ provides no additional information gain regarding $A$. Formally, $P(A|Q, R_{\text{true}}) = P(A|R_{\text{true}})$.

**Theorem (Data Processing Inequality)**: For a Markov chain $X \to Y \to Z$, we have $I(X;Z) \leq I(Y;Z)$.

**Proof**:

1. **Information Transfer**: Applying the Data Processing Inequality to our Markov chain $Q \to R_{\text{true}} \to A$:
$$I(Q;A) \leq I(R_{\text{true}};A)$$

    This means the amount of information transferred to the answer $A$ via the perfect reasoning path $R_{\text{true}}$ is not less than the information transferred directly from the problem $Q$. In an ideal scenario, $I(Q;A) = I(R_{\text{true}};A)$, as $R_{\text{true}}$ acts as a sufficient statistic for $A$ with respect to $Q$.

2. **Conditional Entropy Reduction (Entropy Decrease)**: Using the definition of mutual information $I(X;Y) = H(Y) - H(Y|X)$, we can write:
$$I(R_{\text{true}};A) = H(A) - H(A|R_{\text{true}})$$
$$I(Q;A) = H(A) - H(A|Q)$$

    Substituting these into the inequality $I(Q;A) \leq I(R_{\text{true}};A)$:
$$H(A) - H(A|Q) \leq H(A) - H(A|R_{\text{true}})$$
$$\implies -H(A|Q) \leq -H(A|R_{\text{true}})$$
$$\implies H(A|R_{\text{true}}) \leq H(A|Q)$$

    **Interpretation**: This inequality demonstrates the "entropy decrease." Given an **ideal and true reasoning path $R_{\text{true}}$**, the uncertainty about the answer $H(A|R_{\text{true}})$ is not higher than the uncertainty given only the problem $H(A|Q)$. This signifies that each correct step in the reasoning process provides valuable information, reducing the overall uncertainty about the final answer.

4.2. IMPERFECT EXPLICIT REASONING PATH: CONDITION FOR NET GAIN FROM IMPERFECT REASONING

**Premise**: Here, we aim to derive the formal condition under which an imperfect, noisy reasoning path $R_{\text{gen}}$ is still more beneficial for determining the true answer $A$ than relying solely on the problem $Q$. In other words, we seek the boundary condition for which the explicit reasoning model $M_E$ outperforms the direct model $M_D$. This translates to finding when the uncertainty about the answer, given the generated reasoning, is less than the uncertainty given only the problem:

$$H(A|R_{\text{gen}}) < H(A|Q)$$

This inequality signifies a net reduction in entropy, meaning the reasoning process provides an overall informational gain despite its imperfections.

**Proof**: Our goal is to transform the inequality $H(A|R_{\text{gen}}) < H(A|Q)$ into a more intuitive condition based on mutual information.

1. **Expressing Entropy via Mutual Information**: We use the fundamental relationship $H(X|Y) = H(X) - I(X;Y)$. Applying this to our target inequality, we get:

$$H(A) - I(A; R_{\text{gen}}) < H(A) - I(A; Q)$$

This simplifies to the core requirement for net gain:

$$I(A; R_{\text{gen}}) > I(A; Q)$$

This means the mutual information between the generated reasoning and the true answer must be greater than the mutual information between the problem and the true answer.

2. **Decomposing the Information in the Reasoning Path**: To understand how $I(A; R_{\text{gen}})$ relates to $I(A; Q)$, we use the chain rule for mutual information:

$$I(A; Q, R_{\text{gen}}) = I(A; Q) + I(A; R_{\text{gen}}|Q)$$

This states that the total information about $A$ contained in the pair $(Q, R_{\text{gen}})$ is the information from $Q$ plus the *additional* information provided by $R_{\text{gen}}$ when $Q$ is already known.

3. **Applying the Markov Chain Assumption**: As established in the problem description, the structure of the explicit reasoning model implies a Markov chain $Q \to R_{\text{gen}} \to A$. This assumption is key, as it posits that once the reasoning path $R_{\text{gen}}$ is generated (which typically includes a restatement of $Q$), the original problem $Q$ provides no *further* information about the answer $A$.

   The formal definition of this Markov chain is $P(A|Q, R_{\text{gen}}) = P(A|R_{\text{gen}})$, which is equivalent to the conditional mutual information being zero:

$$I(A; Q|R_{\text{gen}}) = 0$$

4. **Connecting the Pieces**: We can also express the joint information $I(A; Q, R_{\text{gen}})$ using the chain rule in a different order:

$$I(A; Q, R_{\text{gen}}) = I(A; R_{\text{gen}}) + I(A; Q|R_{\text{gen}})$$

Since we established from the Markov assumption that $I(A; Q|R_{\text{gen}}) = 0$, this simplifies to:

$$I(A; Q, R_{\text{gen}}) = I(A; R_{\text{gen}})$$

By equating this with our first chain rule expression from step 2, we get:

$$I(A; R_{\text{gen}}) = I(A; Q) + I(A; R_{\text{gen}}|Q)$$

This identity is central: it beautifully decomposes the information content of the generated reasoning path. It states that the total information in $R_{\text{gen}}$ about $A$ is the sum of the information that was already available in the question ($I(A; Q)$) and the **new, value-added information generated by the reasoning process itself** ($I(A; R_{\text{gen}}|Q)$).

5. **Establishing the Final Condition**: Now, we substitute this decomposition back into our inequality from step 1:

$$I(A; Q) + I(A; R_{\text{gen}}|Q) > I(A; Q)$$

Subtracting $I(A; Q)$ from both sides gives us the final condition for net gain:

$$\mathbf{I(A; R_{\text{gen}}|Q) > 0}$$

**Conclusion**: The proof establishes a clear and intuitive boundary for the effectiveness of explicit reasoning, even when the reasoning path $R_{\text{gen}}$ is imperfect.

- **Condition for Benefit**: An imperfect reasoning path $R_{\text{gen}}$ provides a net gain over direct prediction from $Q$ if and only if the reasoning process itself generates **new information about the answer** $A$ **that was not available from the problem** $Q$ **alone**. $I(A; R_{\text{gen}}|Q)$ quantifies this "value-added" information.

- **The Role of Noise/Error**: The errors and noise introduced by the reasoning generator $P_{\text{path}}$ directly degrade the quality of the reasoning path. This causes the generated path $R_{\text{gen}}$ to be a less faithful representation of the ideal path $R_{\text{true}}$. Consequently, the amount of new information generated is reduced:

$$I(A; R_{\text{gen}}|Q) \leq I(A; R_{\text{true}}|Q)$$

However, as long as this degraded, noisy reasoning process can still uncover *any* amount of new, relevant information about the answer, the value of $I(A; R_{\text{gen}}|Q)$ will be greater than zero, and the overall model will see a benefit ($H(A|R_{\text{gen}}) < H(A|Q)$).

- **The Boundary Case**: The boundary where the explicit reasoning model provides no benefit ($H(A|R_{\text{gen}}) = H(A|Q)$) occurs precisely when $I(A; R_{\text{gen}}|Q) = 0$. This describes a scenario where the entire reasoning process, however lengthy or complex, fails to produce any new insight relevant to finding the answer beyond what was already contained in the question itself.

## 5. CONCLUSION

Based on the detailed mathematical derivation, we conclude:

1. **Ideal Explicit Reasoning Benefit**: In an ideal scenario, if the reasoning path generator perfectly produces the **true, informative reasoning path** $R_{\text{true}}$, the process is **entropy-decreasing**, i.e., $H(A|R_{\text{true}}) \leq H(A|Q)$. A perfect reasoning path effectively reduces the uncertainty about the final answer, theoretically enhancing reliability.

2. **Imperfect Reasoning Net Gain**: In practice, a generated reasoning path ($R_{\text{gen}}$) is imperfect and contains noise. However, this flawed path still provides a net benefit as long as the reasoning process itself uncovers new, value-added information about the answer (A) that was not available from the problem (Q) alone. The formal condition for this benefit is $I(A; R_{\text{gen}}|Q) > 0$. This means even a noisy reasoning process is effective if it successfully reduces the overall uncertainty ($H(A|R_{\text{gen}}) < H(A|Q)$), outperforming a direct prediction. The benefit disappears only when the reasoning provides no new information ($I(A; R_{\text{gen}}|Q) = 0$).

## C   EXTENSIVE EXPERIMENTS & ANALYSIS

### C.1   IMPACT OF REASONING WITH DATA QUANTITY ON COLD-START PERFORMANCE

Figure 6 investigates the impact of varying data quantities on model performance in cold-start scenarios, specifically comparing models with and without reasoning capabilities.

#### C.1.1   LOW DATA SCENARIO

In the initial cold-start SFT step, where data is extremely limited, models incorporating reasoning capabilities demonstrate a decisive advantage. Both in terms of Levenshtein Distance (lower values

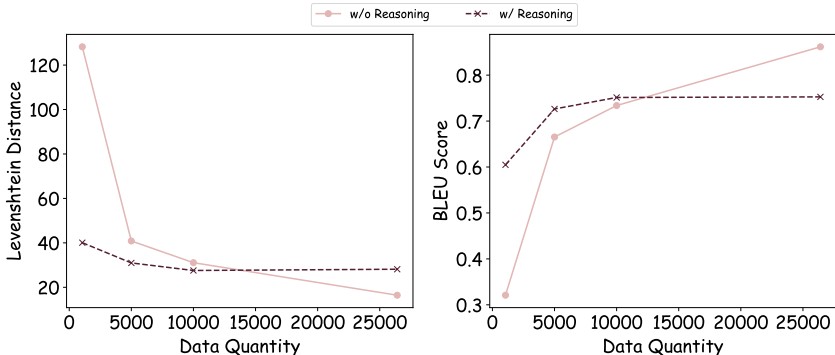

Figure 6: Impact of reasoning on the cold-start supervised fine-tuning performance of LLMs in Text-based Molecule Generation with different data quantities. Liechtenstein Distance (Left); BLEU Score (Right).

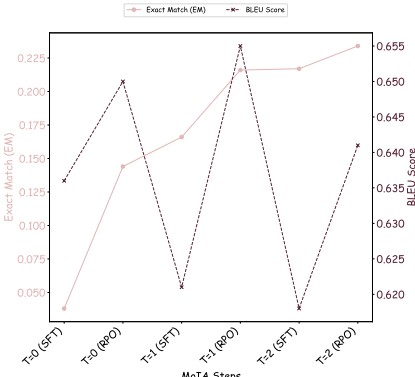

Figure 7: The performance change across different MoIA steps and iterations.

indicate better performance) and BLEU Score (higher values indicate better quality), reasoning models significantly outperform their counterparts without reasoning. This suggests that in an information-poor environment, incorporating reasoning traces enables models to extract deeper patterns and relationships from limited data, leading to more accurate and expected outputs.

### C.1.2 PERFORMANCE WITH INCREASING DATA QUANTITY

As the available data quantity gradually increases, both model types exhibit positive performance improvements. Initially, models without reasoning traces rapidly improve, and with sufficient data, they can even surpass reasoning models in certain metrics. This phenomenon can be attributed to the nature of reasoning traces distilled from LLMs. As previously discussed, these traces represent an imperfect explicit reasoning path. When the inherent noise within these distilled traces outweighs the information gains they provide, their presence can actually hinder performance compared to directly generating final outputs.

### C.2 MODEL PERFORMANCE ACROSS MoIA STEPS AND ITERATIONS

Table 4 and Figure 7 present a comprehensive comparison of model performance across different MoIA iterations (T=0, T=1, T=2) and middle steps in MoIA: Supervised Fine-Tuning (SFT) and Reasoning Policy Optimization (RPO).

Overall, the results consistently highlight the effectiveness of the RPO within each MoIA iteration. RPO generally enables models to outperform their SFT counterparts at each MoIA iteration, indicated by higher exact match (EM) and Molecule FTS scores.

| Method | BLEU↑ | EM↑ | Levenshtein↓ | MACCS FTS↑ | RDK FTS↑ | Morgan FTS↑ | Validity↑ |
|---|---|---|---|---|---|---|---|
| **MoIA T=0** | | | | | | | |
| T=0 (SFT) | 0.636 | 0.038 | 39.67 | 0.761 | 0.549 | 0.475 | 0.750 |
| T=0 (RPO) | 0.650 | 0.144 | 34.33 | 0.804 | 0.639 | 0.558 | 0.819 |
| **MoIA T=1** | | | | | | | |
| T=1 (SFT) | 0.621 | 0.166 | 34.22 | 0.802 | 0.640 | 0.561 | 0.848 |
| T=1 (RPO) | 0.655 | 0.216 | 32.87 | 0.821 | 0.673 | 0.596 | 0.863 |
| **MoIA T=2** | | | | | | | |
| T=2 (SFT) | 0.618 | 0.217 | 34.06 | 0.804 | 0.658 | 0.583 | 0.879 |
| T=2 (RPO) | 0.641 | 0.234 | 32.94 | 0.823 | 0.684 | 0.612 | 0.847 |

Table 4: Comparative performance of models at different MoIA iterations and middle steps (SFT, RPO). The best and second-best results are highlighted in red and blue , respectively.

Meanwhile, we observe a consistent increase in exact match (EM) scores during the three iterations of MoIA training, as shown in Figure 7. The initial EM is quite low at T=0's SFT step, only 0.038. However, by implementing RPO at T=0, the EM score significantly jumps to 0.144, demonstrating the immediate positive impact of policy optimization. As the training progresses to T=1, both SFT and RPO further improve, with RPO again showing a stronger result of 0.216 compared to SFT's 0.166. This upward trend continues into T=2, where the EM scores reach their highest points: SFT achieves 0.217, and RPO culminates with the best overall EM score of 0.234. This steady rise in EM across iterations underscores MoIA's effectiveness in refining the model's ability to generate precisely correct outputs, with RPO consistently leading in this crucial metric.

However, we also observe an interesting trend: the BLEU score exhibits a fluctuating pattern, showing no significant monotonic improvement across the MoIA iterations. This behavior, particularly as the EM score converges, seems to indicate an upper bound on the model's capability. This suggests a performance ceiling reachable through self-learned policies under the constraints of the limited and potentially imperfect reasoning traces used for annotation. Ultimately, we hypothesize that this ceiling is determined by an interplay of the model's inherent capacity and the quality of the initial cold-start reasoning data.

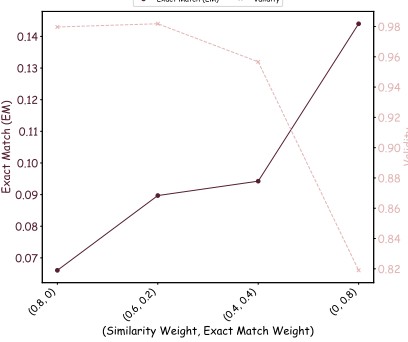

Figure 8: The performance change with different weights of similarity and exact match in the reward model.

### C.3 IMPACT OF REWARD MODELLING

Figure 8 illustrates the model's performance across various reward function combinations. Here, we specifically examine the interplay between Exact Match and Validity metrics. Notably, the weights for the similarity and Exact Match rewards sum to 0.8; the remaining 0.2 is allocated to the length reward.

We observe a clear inverse relationship between Exact Match and Validity as the weighting shifts. As the Exact Match weight increases (moving from left to right on the x-axis), the Exact Match score

also consistently rises. This indicates that placing more emphasis on exact matches in the reward function effectively drives the model to produce more precise outputs.

Conversely, as the Exact Match weight increases, the Validity score consistently declines. This suggests a trade-off: optimizing too aggressively for exact matches leads to a degradation in the structural integrity or chemical correctness of the generated outputs. The initial configurations, where the Similarity weight is higher, maintain very high Validity, highlighting its importance in preserving output correctness.

In essence, the trend reveals that while boosting Exact Match score is achievable by adjusting the reward function, it often comes at the cost of reduced Validity. However, in this work, we prioritize the Exact Match score more heavily because achieving a precise and chemically correct match to the target is paramount for the specific downstream applications this model is designed for.

## D    EXAMPLES OF REASONING TRACES GENERATED BY GPT-4O VIA PRID

Here, we demonstrate two examples of the reasoning traces that are distilled from GPT-4o via PRID, as demonstrated in Figures 9 and 10. The reasoning traces show clearly how to infer detailed molecular structures based on the textual representations. Basically, the reasoning traces identify keywords in the natural language description, use them to form a molecular structure hypothesis, and then refine the hypothesis by considering additional details in the description, such as functional groups, their positions, and the overall molecular class.

---

**Caption:** The molecule is a 2-hydroxydicarboxylic acid that is succinic acid in which one of the hydrogens attached to a carbon is replaced by a hydroxy group. It has a role as a food acidity regulator and a fundamental metabolite. It is a 2-hydroxydicarboxylic acid and a C4-dicarboxylic acid. It derives from a succinic acid. It is a conjugate acid of a malate(2-) and a malate.

**Molecule:** C(C(C(=O)O)O)C(=O)O

**Reasoning Process from Caption to Molecule:**
<think>
😊 Well, I now need to generate a SMILES representation for a molecule based on the description provided by the user. First, I have to carefully analyze each sentence to understand the structural features they correspond to.
First sentence: "The molecule is a 2-hydroxydicarboxylic acid that is succinic acid in which one of the hydrogens attached to a carbon is replaced by a hydroxy group." There are a few keywords here: 2-hydroxydicarboxylic acid, succinic acid, and hydroxy group. 🤔 Let me think about what the SMILES for succinic acid looks like.
🤔 Succinic acid is a four-carbon dicarboxylic acid, and its SMILES representation is C(CC(=O)O)C(=O)O. According to the description, we need to replace one of the hydrogen atoms on a carbon with a hydroxy group. Based on the "2-hydroxy" description, the hydroxy group should be on the second carbon. Therefore, the SMILES representation for this molecule should be C(C(C(=O)O)O)C(=O)O.
Second sentence: "It has a role as a food acidity regulator and a fundamental metabolite." This part mainly describes the function and properties of the molecule. 🤔 Let me think, food acidity regulators and fundamental metabolites, these functions are closely related to the molecule's structure. Food acidity regulators are usually organic acids, while fundamental metabolites are often important compounds involved in cellular metabolic processes. This molecule is a dicarboxylic acid with an added hydroxy group, which fits these properties.
Third sentence: "It is a 2-hydroxydicarboxylic acid and a C4-dicarboxylic acid." This confirms the structural features of the molecule again: a 2-hydroxydicarboxylic acid and a C4-dicarboxylic acid. C4-dicarboxylic acid means the molecule has four carbon atoms and two carboxyl groups.
Fourth sentence: "It derives from a succinic acid." This mentions that the molecule is derived from succinic acid, confirming our previous inference again.
Fifth sentence: "It is a conjugate acid of a malate(2-) and a malate." This mentions that the molecule is the conjugate acid of malate. Malate is a C4-dicarboxylic acid, a dicarboxylic acid plus a hydroxy group, which matches our previous deduction.
Now, based on my current line of thought, I need to write out the SMILES representation for the molecule. The SMILES representation for this molecule should be: C(C(C(=O)O)O)C(=O)O.
This SMILES representation meets our expectations; it is derived from succinic acid and also fits the characteristics of a 2-hydroxydicarboxylic acid. 🎉
Next, I need to output this SMILES according to the format specified by the user.
</think>

---

Figure 9: The reasoning process generated by GPT-4o via PRID. Here, this example demonstrates how to obtain the target molecule C(C(C(=O)O)O)C(=O)O based on the content of the molecule caption.

## E    PROMPT DESIGN

In this section, we demonstrate the prompts we apply in this work. Figure 11 illustrates the details of the expert-annotated context example for PRID. This example provides a clear and detailed reasoning trace that guides the LLM through the process of generating a molecular structure from a textual description. The reasoning trace serves as a form of "scaffolding" that regulates the LLM's exploration behavior by breaking down a complex problem into a series of manageable steps.

### E.1    EXPERT-ANNOTATED CONTEXT EXAMPLE FOR PRID

The process unfolds as follows:

Caption: The molecule is the *C*-nitro compound that is chlorobenzene with three nitro substituents in the 2-, 4- and 6-positions. It has a role as an epitope, an explosive, a hapten and an allergen. It is a *C*-nitro compound and a member of monochlorobenzenes.

Molecule: C1=C(C=C(C(=C1[N+](=O)[O-])Cl)[N+](=O)[O-])[N+](=O)[O-]

Reasoning Process from Caption to Molecule:
<think>
😊 Well, I now need to generate a SMILES representation for a molecule based on the description provided by the user. First, I have to carefully analyze each sentence to understand the structural features they correspond to.
First sentence: "The molecule is the *C*-nitro compound that is chlorobenzene with three nitro substituents in the 2-, 4- and 6-positions." There are several keywords here: *C*-nitro compound, chlorobenzene, three nitro substituents in the 2-, 4- and 6-positions. 🤔 Let me think about what the SMILES for chlorobenzene is.
The SMILES for chlorobenzene is: ClC1=CC=CC=C1. This is a benzene ring where one hydrogen atom is replaced by a chlorine atom. Next, according to the description, this chlorobenzene has three nitro substituents at positions 2, 4, and 6. We need to add nitro groups (-NO2) at these appropriate positions. 🤔 The SMILES for a nitro group is [N+](=O)[O-]. So we need to add nitro groups to chlorobenzene at positions 2, 4, and 6 respectively.
Second sentence: "It has a role as an epitope, an explosive, a hapten and an allergen." This part mainly describes the molecule's functions and properties. 🤔 Let me think about this: an epitope is a specific region on an antigen molecule that can be recognized by antibodies. An explosive indicates this molecule has high energy and instability. A hapten is a small molecule that cannot elicit an immune response alone but can do so when conjugated with a carrier protein. An allergen is a substance capable of causing allergic reactions.
These functions and properties suggest the molecule has significant chemical reactivity and instability, likely related to its multiple nitro groups (-NO2), as nitro is a strong electron-withdrawing group that increases molecular reactivity and instability.
Third sentence: "It is a *C*-nitro compound and a member of monochlorobenzenes." Here it's mentioned that the molecule is a *C*-nitro compound and belongs to the monochlorobenzene family. Being a *C*-nitro compound means it contains nitro groups (-NO2), and being a monochlorobenzene member indicates it contains one chlorine atom.
Based on these descriptions, we can determine the molecule's basic structure is chlorobenzene (ClC1=CC=CC=C1) with nitro groups (-NO2) at positions 2, 4, and 6. We can generate the final SMILES representation by adding nitro groups at these appropriate positions.
Now, we need to write the SMILES representation based on our current reasoning. First, we write the chlorobenzene structure: ClC1=CC=CC=C1. Then, we add nitro groups (-NO2) at positions 2, 4, and 6 respectively to obtain the final SMILES: C1=C(C=C(C(=C1[N+](=O)[O-])Cl)[N+](=O)[O-])[N+](=O)[O-].
This SMILES representation meets our expectations; it's a chlorobenzene with nitro groups (-NO2) at positions 2, 4, and 6, and also aligns with the characteristics of a *C*-nitro compound and monochlorobenzene. 🎉
Next, we need to output this SMILES according to the format specified by the user.
</think>

Figure 10: The reasoning process generated by GPT-4o via PRID. Here, this example demonstrates how to obtain the target molecule `C1=C(C=C(C(=C1[N+](=O)[O-])Cl)[N+](=O)[O-])[N+](=O)[O-]` based on the content of the molecule caption.

- **Decomposition of the Problem**: The LLM is first prompted to analyze the user's caption sentence by sentence, which encourages it to identify keywords and extract specific structural information from each piece of text. For instance, it identifies "cyclodiene," "organochlorine," and "indene derivative." This step prevents the LLM from attempting to solve the entire problem at once and instead focuses its attention on individual, verifiable facts.

- **Iterative Hypothesis Formulation and Refinement**: The LLM is guided to build a hypothesis incrementally. It starts with the initial, vague keyword "cyclodiene" and considers several possibilities (cyclopentadiene, cyclohexadiene). When this is not enough, it is instructed to look for clues in subsequent sentences. The mention of "indene derivative" is a crucial turning point, allowing the LLM to narrow down its initial hypothesis to a structure derived from indene. This demonstrates how the reasoning trace directs the LLM to seek out and integrate new information to refine its understanding, moving from a general concept to a more specific structural scaffold.

- **Connecting Function to Structure**: The context example prompts the LLM to consider functional descriptions like "GABA-gated chloride channel antagonist" and "persistent organic pollutant." While these properties don't directly define the structure, they provide contextual clues that help the LLM make informed decisions. For example, recognizing that the molecule is a persistent organochlorine insecticide with multiple chlorine atoms strengthens the hypothesis of a complex, chlorinated hydrocarbon structure, moving it beyond simple chlorinated indene. This step guides the LLM to validate its structural assumptions against the molecule's known properties.

- **Structured Generation and Verification**: Finally, the LLM is guided to synthesize all the information to generate the final SMILES representation. It is instructed to first think about the core structure (the indene derivative), then consider the modifications (chlorination and the elimination of double bonds), and finally, to verify that the generated SMILES aligns with all parts of the initial description. This structured approach ensures that the final output is not a guess but a well-reasoned conclusion derived from a systematic process of information synthesis and validation.

In essence, the expert-annotated context example in PRID serves as a detailed blueprint for problem-solving. It demonstrates a logical and verifiable path from textual description to molecular structure,

thereby providing a clear and effective form of guidance that regulates the LLM's reasoning process and improves the accuracy of its generated outputs.

---

**Caption:** The molecule is a cyclodiene organochlorine insecticide. It has a role as a GABA-gated chloride channel antagonist and a persistent organic pollutant. It derives from a hydride of an indene.

**Molecule:** C1C2C(C(C1Cl)Cl)C3(C(=C(C2(C3(Cl)Cl)Cl)Cl)Cl)Cl

**Reasoning Process from Caption to Molecule:**
\<think\>
😊 Okay, I'm now tasked with generating the SMILES representation of a molecule based on the user's description. First, I need to carefully analyze each sentence and understand the corresponding structural features.
The first sentence: "The molecule is a cyclodiene organochlorine insecticide." There are several keywords here: cyclodiene, organochlorine, and insecticide. 😊 Let me think, what does a cyclodiene SMILES look like?
😊 There seem to be several types of cyclodienes, differentiated by the number of carbon atoms, such as cyclopentadiene (C1=CC=CC1) and cyclohexadiene (C1=CCC=CC1, C1C=CC=CC1). However, the description only mentions cyclodiene, and I'm currently unsure which one it is.
Well, I can look for information elsewhere. According to the second keyword, this molecule is an organochlorine compound, right? So, there must be chlorine atoms in the molecule's SMILES! Well, a common organochlorine compound is chloroform. Its SMILES is ClC(Cl)Cl. There are several chlorine atoms, which suggests that this molecule might also have several chlorine atoms. But this is still not enough to deduce the entire structure of the molecule.
Well, what does the third keyword tell me? Oh, this molecule is an insecticide. Let me think, an insecticide containing chlorine, right? Oh, dichlorodiphenyltrichloroethane (DDT) is a common chlorine-containing insecticide, and its SMILES is ClC(Cl)(Cl)C1=CC=C(Cl)C(Cl)=C1. 😊 It looks very complicated, and I still don't have any clues. Maybe I should look at the second sentence.
The second sentence: "It has a role as a GABA-gated chloride channel antagonist and a persistent organic pollutant." This part mainly describes the function and properties of the molecule.
😊 Let me think,, first of all, a GABA-gated chloride channel antagonist, indicating that this is a type of drug that affects the central nervous system. GABA is the main inhibitory neurotransmitter in the central nervous system, and the GABA-gated chloride channel is part of the GABA receptor. When GABA binds to the receptor, the chloride channel opens, chloride ions flow into the neuron, causing hyperpolarization of the neuron, thereby inhibiting the excitability of the neuron. So, such a molecule needs to have certain properties, which may be reflected in the structural features of the molecule. For example: having specific binding sites that can bind to specific regions of the GABA receptor, thereby preventing the action of GABA. 😊 This property is a bit difficult to analyze. I only know that halogens may affect its pharmacological activity, but I'm not sure which points on the molecule. However, based on other properties, this molecule also needs to have a certain lipophilicity and electroneutrality, indicating that this molecule is usually hydrophobic and needs to exclude some hydrophilic groups. In addition, this molecule should also have no ions.
Furthermore, this molecule is a persistent organic pollutant. Wait, this is very similar to DDT, right! 😊 But organochlorine compounds seem to have this characteristic.
The third sentence: "It derives from a hydride of an indene." Here, indene is mentioned, and the molecule I need to generate is its derivative. Indene is a bicyclic structure consisting of a benzene ring and a cyclopentene. As a hydride, it means that a proton has been removed from indene, or perhaps a certain group has been added at a certain position.
😊 So I need to determine what the SMILES of indene is. Well, as far as I know, this is an aromatic hydrocarbon with the molecular formula C6H4C3H4, which can be regarded as a benzene ring fused with a cyclopentadiene, and the SMILES can be written as C1C=Cc2ccccc21, or C1C=CC2=CC=CC=C21. 😊 I understand, the cyclodiene in the first sentence is actually cyclopentadiene! 😊 But I need to consider whether this molecule is a simple chlorinated indene. If so, the SMILES of this molecule would be very simple.
However, the description in the first sentence emphasizes cyclodiene, and the second sentence emphasizes its antagonist role, so the molecule may not be so simple. However, in any case, the final molecular structure should be derivable from indene.
Wait, I remember that there is a cyclopentadiene insecticide that usually has a structure, which is formed by two cyclopentadiene rings sharing three carbon atoms. In this way, if the central carbon atom is removed, the remaining structure will be a six-membered ring! As long as an additional cyclopentadiene is added next to it, it can be ensured that the molecule is derived from indene!
Now, I need to write the SMILES representation of the molecule based on the current ideas. First, I need to write the structure before chlorination, then eliminate some double bonds and add chlorine atoms. Finally, I need to check whether the molecule meets our expectations.
😊 The structure before chlorination consists of three cyclopentadiene rings, two of which share three carbon atoms due to the three carbon atoms, making six carbon atoms aromatic, and a single carbon atom detached from the benzene plane. The SMILES representation of this structure can be written as C1C=CC2=C3C=CC(C3)=C21, right!
Next, I will eliminate double bonds and add chlorine atoms. However, depending on the number of chlorine atoms, I can get multiple molecules that meet the description. Here, I need to combine my speculation about the other properties of this molecule, such as my understanding of chlorinated insecticides, to determine the final SMILES representation.
Based on the structure of common chlorinated insecticides, I find that these compounds have multiple chlorine atoms, and their chemical properties require the presence of double bonds. Therefore, I believe that the final SMILES representation of this molecule is: C1C(Cl)C(Cl)C2C3(Cl)C(Cl)=C(Cl)C(Cl)(C3(Cl)Cl)C21, well 😊, let me summarize, this molecule can also be written as: C1C2C(C(C1Cl)Cl)C3(C(=C(C2(C3(Cl)Cl)Cl)Cl)Cl)Cl.
This SMILES representation meets our expectations, it is derived from indene, and also meets the characteristics of organochlorine insecticides. 🎉
Next, I need to output this SMILES according to the format specified by the user.
\</think\>
\<answer\>
C1C2C(C(C1Cl)Cl)C3(C(=C(C2(C3(Cl)Cl)Cl)Cl)Cl)Cl
\</answer\>

---

Figure 11: The expert-annotated context example for Prior Regulation via In-context Distillation.

## E.2 PROMPT FOR EVALUATING CONSISTENT-F1

Finally, Figure 12 shows the prompt for evaluating the Consistent-F1 score, which aims to reveal the reasoning traces quality for explicit Long-CoT reasoning models.

# F USE OF LLMS

During the preparation of this work, the author(s) used LLMs to improve the language and readability. After using this tool/service, the author(s) reviewed and edited the content as needed and take(s) full responsibility for the content of the publication.

You are an excellent chemist and an AI assistant for evaluating reasoning traces.

Here is a text-based molecule generation problem, which aims to predict the SMILES representation of the molecule based on the molecular description.

Your assistant has provided a detailed reasoning trace based on the molecular description.
Your task is to examine the reasoning trace and determine whether it could correctly predict the SMILES representation of the molecule.
Your judgment should be based on the factual correctness and logic coherence of the reasoning trace.

Here is the original molecular description:
{description}
Here is the reasoning trace from your assistant:
{reasoning_trace}

Your response should be 'yes' if the reasoning trace is consistent with the molecular description and could lead to a correct SMILES representation, or 'no' if it is not consistent.

Your response should be in the following format:
{
"response": "yes" or "no"
}

YOUR OUTPUT SHOULD BE A VALID JSON OBJECT. YOU MUST NOT INCLUDE ANY ADDITIONAL TEXT OR COMMENTS.

Figure 12: The prompt for evaluating the quality of reasoning traces (i.e., **Consistent-F1**).

