# OpenReview forum: "Mol-R1: Towards Explicit Long-CoT Reasoning in Molecule Discovery"
_ICLR.cc/2026/Conference — ICLR 2026 Conference Withdrawn Submission_

### Official Review · Reviewer_uAbq · 2025-10-27

**Soundness:** 2
**Presentation:** 2
**Contribution:** 2
**Rating:** 2
**Confidence:** 5

**Summary:**

This paper proposes Mol-R1, a framework for generating molecules from text descriptions using reasoning traces.
The approach consists of two components: (1) Prior Regulation via In-context Distillation (PRID), which generates initial reasoning traces by prompting GPT-4o with an expert-annotated example and followed filtering, and (2) Molecular Iterative Adaptation (MoIA), which iteratively refines the model through cycles of supervised fine-tuning, reinforcement learning, and rejection re-sampling.
The authors evaluate on the ChEBI-20 dataset and report improvements in exact match scores and molecular fingerprint metrics compared to baseline models including DeepSeek-R1.

**Strengths:**

- Important problem. Explainability in molecule generation for drug discovery is genuinely valuable, and the motivation for transparent reasoning is well-articulated.
- The experimental evaluation is reasonably comprehensive in scope. The authors compare against multiple strong baselines, including recent reasoning models like QWQ-32B and DeepSeek-R1, as well as GPT-OSS-120b.
- Multiple metrics beyond accuracy (fingerprint similarity, validity measures)
- The case study in Figure 5 provides useful qualitative insight by showing how reasoning traces evolve across training iterations, demonstrating that the model progressively refines its understanding of molecular structure.

**Weaknesses:**

- **Unvalidated reasoning claims.** Title emphasizes "Explicit Long-CoT Reasoning," but Mol-R1 produces ~428-word traces versus 5,337 (QWQ-32B) and 4,518 (DeepSeek-R1)—actually *shorter*, not longer. No analysis of: (1) whether reasoning length correlates with performance, (2) whether traces causally contribute versus merely correlate, (3) generalization to novel molecules.The claimed novelty misrepresents standard techniques.
 - **Misrepresented novelty.** PRID (Figure 11) is standard one-shot prompting of GPT-4o, obscured by terminology. MoIA exactly follows STaR [Zelikman et al., 2022]—iteratively generating traces, filtering correct ones, retraining—and expert iteration [Anthony et al., 2017; Silver et al., 2017], yet neither is cited. Related work omits these foundational papers entirely.
 - **Missing critical ablations.** Need: (1) matched-budget comparison of models trained with/without reasoning (current ablations lack base model trained without reasoning at same budget), (2) reasoning length vs. performance analysis, (3) whether improvements come from reasoning versus more training data. No evaluation of generalization to distribution shifts or novel scaffolds.
 - **Flawed reasoning evaluation.** Consistent-F1 uses Gemma-3-27B as judge without validation—no inter-rater reliability with chemists, no multi-judge comparison. The metric appears to include all traces including failed predictions, mixing signal with noise when it should condition only on successful predictions. Figure 3 shows Mol-R1-RS-G achieves Consistent-F1 ~0.25, beating some MoIA iterations, yet this is never discussed. Additionally, an ablation where reasoning traces are paired with separate but correct SMILES strings would be highly valuable to assess the judge's discernment—if judgment agreement remains high despite mismatched reasoning-molecule pairs, the judge cannot distinguish reasoning quality from answer correctness, invalidating the metric.
 - **Missing experimental details.** Table 1: no error bars, inference procedures, or computational costs. BLEU questionable for SMILES—may just measure canonical notation learning. Mol-R1-RS-G exact match vs. fingerprint discrepancy unexplained. Figure 4 "inference" curve undefined; train-inference gap not discussed.
 - **Incomplete related work.** Section 2.2 ignores ChemBERTa, MolGPT, and extensive prior work. Foundational reasoning papers (STaR, ReAct) uncited. Line 102 claim that "parameter scale creates reasoning capability" is misleading—o1 and DeepSeek-R1 success comes from RL on top of capable base models, not scale alone.
 - **Unsubstantiated claims.** Lines 64-65: reasoning enables "human-interpretable justifications" without human evaluation from chemists. Lines 216-217: existing LLMs "fail due to lack of molecular knowledge" without supporting experiments, contradicting successful deployments.

### Additional References
 - Eric Zelikman, Yuhuai Wu, and Noah D Goodman. Star: Self-taught reasoner. In Proceedingsof the NIPS, volume 22, 2022.
 - Thomas Anthony, Zheng Tian, and David Barber. Thinking fast and slow with deep learningand tree search. Advances in neural information processing systems, 30, 2017.
 - David Silver, Julian Schrittwieser, Karen Simonyan, Ioannis Antonoglou, Aja Huang, ArthurGuez, Thomas Hubert, Lucas Baker, Matthew Lai, Adrian Bolton, et al. Mastering the gameof go without human knowledge. nature, 550(7676):354–359, 2017.

**Questions:**

1. Can you provide ablations training models with/without reasoning on matched-quality data, controlling for everything except reasoning text presence?
2. How do you justify "long-CoT" when traces are 1/10th baseline length? Does reasoning length correlate with performance?
3. Can you evaluate on molecules created after GPT-4o and Llama-3.1-8B training cutoffs to rule out contamination?
4. How is Consistent-F1 computed? Do you include failed predictions? Have you validated Gemma-3-27B as a qualified judge against human chemists?
5. Why does Mol-R1-RS-G beat MoIA iterations on Consistent-F1 (Figure 3)?
6. What distinguishes your approach from STaR [Zelikman et al., 2022] and expert iteration [Anthony et al., 2017]?
7. Can you provide human evaluation from chemists on reasoning interpretability and trustworthiness?
8. The BLEU scores show interesting patterns—they fluctuate across MoIA iterations rather than monotonically improving, even as exact match scores improve. What explains this discrepancy? Is BLEU actually measuring something meaningful for SMILES strings, or is it primarily capturing whether the model has learned the canonical notation format?

**Details Of Ethics Concerns:**

- **Interpretability risk** Without human‑chemist validation, traces may be plausible but unfaithful, inflating trust.
 - **Responsible release** If releasing models/traces, include misuse policies and filters for hazardous chemistry prompts.

---

### Official Review · Reviewer_Rokw · 2025-10-31

**Soundness:** 2
**Presentation:** 2
**Contribution:** 2
**Rating:** 2
**Confidence:** 5

**Summary:**

This paper proposed Mol-R1, a framework to improve explainability of molecule discovery with LLMs. It presented techniques including PRID and MoIA for training the LLM to deal with pre-regulation and post-regulation problems.

**Strengths:**

- The figure effectively increases the understandability of the paper.
- The paper proposes RL fine-tuning reasoning approach for molecular generation.

**Weaknesses:**

- **Lack of baselines and details of experiment**: It is unclear which dataset was used in Table 1. From the description, it seems to be based on the *ChEBI-20* dataset, but the experimental settings are not explicitly stated. If *ChEBI-20* was indeed used, the authors should have compared against established caption-based molecular generation baselines such as MolT5 [1], BioT5 [2], and MSR [3]. Moreover, the reported metrics appear substantially lower than those achieved by these baselines, which raises concerns about the experimental results.
- **Complexity of terms**: The paper introduces several new terms without clear justification. For example, the definition of PRID is vague—does it merely concatenate expert-annotated contexts with raw training data before querying the LLM? The claimed novelty of the PRID method remains unclear. In addition, although the authors adopt GRPO as their reinforcement learning strategy, they refer to their approach as RPO (only maximizing the reward without KL penalty, which causes unnecessary confusion.
- **Limited applicability beyond text-based molecular generation task**: The *ChEBI-20* text-based generation task is arguably unrealistic, as many captions explicitly include IUPAC names or overly detailed structural descriptions, making direct molecule reconstruction trivial or uninformative. It would be valuable to demonstrate the proposed framework’s generality on other, more practical tasks.



[1] Edwards et al., Translation between Molecules and Natural Language, EMNLP 2022.

[2] Pei et al., BioT5: Enriching Cross-modal Integration in Biology with Chemical Knowledge and Natural Language Associations, EMNLP 2023.

[3] Jang et al., Structural Reasoning Improves Molecular Understanding of LLM, ACL 2025

**Questions:**

- **Model choice for baselines**: Why did the authors exclusively use Gemma models for rejection sampling? For a fair comparison, shouldn’t the same base model be used across settings? Furthermore, the paper lacks justification for using multiple model sizes—Gemma-27B, Gemma-12B, and Llama-3.1-8B—in different parts of the experiment.
- **Reward sparsity in GRPO**: Isn’t EM reward too sparse to train GRPO? Have the authors experimented with alternative or complementary reward signals, such as similarity-based metrics?

---

### Official Review · Reviewer_27sz · 2025-10-31

**Soundness:** 3
**Presentation:** 3
**Contribution:** 3
**Rating:** 6
**Confidence:** 4

**Summary:**

Mol-R1 introduces a reasoning-first framework for molecule generation using natural language descriptions. It generates explicit reasoning traces before outputting a SMILES molecule string. The training pipeline combines reasoning trace distillation from GPT-4 (PRID) and a stable supervised + RL training loop (MoIA). Experiments show strong improvements in generation accuracy and reasoning consistency over existing methods.s

**Strengths:**

1. The model generates high-quality, interpretable reasoning traces, and outperforms larger models on molecule generation accuracy

2. Mol-R1 presents an innovative reasoning-trace data generation via PRID

3. The works enables stable learning through MoIA’s alternating SL and RL

4. The authors defines a consistency metric between reasoning and output

**Weaknesses:**

1. The reasoning quality is judged by an automated LLM, not human experts

2. The paper only tests the Mol-R1 model on one specific task: generating a SMILES molecule from a natural language description, i.e. text-to-SMILES. It leaves open the question of how well the model would perform on other types of molecular design tasks, such as molecule optimization for certain properties.

**Questions:**

1. Does PRID work effectively with weaker teacher models?

2. How does Mol-R1 generalize to broader molecule design tasks?

3. The paper uses LLM to score how consistent the model’s reasoning is with the output molecule (Consistent-F1). Has the authors conducted any human expert evaluation to verify that the reasoning is actually chemically sound?

---

### Official Review · Reviewer_R386 · 2025-11-01

**Soundness:** 2
**Presentation:** 2
**Contribution:** 3
**Rating:** 4
**Confidence:** 4

**Summary:**

The authors propose a training pipeline for chemical reasoning models based on expert iterations. The SFT data is generated using one-shot in-context distillation, where the single demonstration consists of a human-annotated reasoning trace. In benchmark experiments, the authors demonstrate the effectiveness of their approach, showing that it outperforms baseline models and consistently improves across successive expert-iteration rounds.

**Strengths:**

* **(S1 - relevance, novelty) - High-quality SFT data as a valuable contribution to the chemical reasoning community.** In-context distillation is a concept well established in the general reasoning domain, and the authors successfully adapt this idea to chemical reasoning scenarios. This adaptation introduces a degree of novelty, particularly because high-quality SFT data is valuable for the community (however, see (W1)).

* **(S2 - relevance, novelty) - Expert iterations demonstrate effectiveness.** Similar to (S1), applying a well-known principle to chemical reasoning tasks is interesting, especially since the authors are able to demonstrate its effectiveness by comparing expert iterations to vanilla GRPO (Figure 4) (however, see (W1)).

**Weaknesses:**

* **(W1 - clarity, quality) - Lack of related work.** The concepts of in-context distillation and expert iterations are well established in the general reasoning community. The authors should elaborate on this in the related work section and clearly state that their main contribution lies in applying these existing concepts to chemical reasoning scenarios.

* **(W2 - quality, significance) - Missing error bars.** All results are presented without error bars or statistical tests. Consequently, it remains unclear to what extent the observed performance gains may have occurred by chance. Since rerunning training might be too costly, the authors could at least report error bars across tasks or samples.

* **(W3 - clarity, minor issue).** At several points, the manuscript’s clarity could be improved:
    - The terms *pre-regulation* and *post-regulation* are not standard. The authors should consider replacing them or providing clear definitions.
    - l144ff:
      > "In contrast, our work introduces text-based molecule reasoning generation, instead of simply translating a molecule into a predefined description."

      This statement is misleading, as prior work has already explored this direction [1].
    - Equation (2) is written in an overly complicated manner. Essentially, the loss corresponds to a standard cross-entropy objective in a next-token prediction setup, where trace and answer tokens do not need to be distinguished.

* **(W4 - clarity, quality) - Lack of information regarding distilled traces.** In-context distillation is performed to improve the quality of the SFT data. However, it remains unclear whether this procedure truly improves data quality (see questions).

**Questions:**

* Could it be demonstrated that in-context distilled samples are of higher quality compared to standard rejection-sampled data?
* How diverse are the in-context distilled samples? Are they content-wise close to the expert example? If so, wouldn’t this pose a potential issue? It seems reasonable to assume that different samples may require distinct reasoning strategies.

### Other comments
* The entropy consideration in Appendix B is correct; however, the implicit conclusion (not explicitly written in the manuscript) that reasoning traces cannot harm but only help is inaccurate. While it is true that increasing the context window allows the model to access more information through the attention mechanism (and never less), this does not imply that additional tokens cannot have negative effects. Intuitively, the model indeed attends to all tokens within the context window, and the entropy argument provides a formal proof of increased accessible information. Nevertheless, intermediate tokens — such as reasoning trace tokens — can still be harmful. For example, imagine a case where the model is provided with a correct molecule in the question but then exposed to an incorrect molecule repeated 100 times within the reasoning trace, making the model effectively forget the original molecule. Admittedly, this example is extreme and constructed, but the same principle can apply to more subtle cases that may arise within real reasoning traces.

* The authors mention at several points (e.g., l146) that reasoning traces ensure interpretability. However, this would only hold if the reasoning traces consistently reflected the model’s actual decision-making process. Since this cannot be guaranteed — and numerous counterexamples have been documented — the claim should be stated with greater caution.

### References
[1] Narayanan. Training a Scientific Reasoning Model for Chemistry.

---

### Note · Authors · 2025-12-29

I have read and agree with the venue's withdrawal policy on behalf of myself and my co-authors.